# TPV Foaming by CO_2_ Extrusion: Processing and Modelling

**DOI:** 10.3390/polym14214513

**Published:** 2022-10-25

**Authors:** Benoit Rainglet, Paul Besognet, Cyril Benoit, Karim Delage, Véronique Bounor-Legaré, Charlène Forest, Philippe Cassagnau, Yvan Chalamet

**Affiliations:** 1Ingénierie des Matériaux Polymères (IMP), Université Jean Monnet Saint-Etienne, Univ-Lyon, CNRS, UMR 5223, 23 rue Dr. P. Michelon, CEDEX, 42023 Saint-Etienne, France; 2Ingénierie des Matériaux Polymères, Université Claude Bernard Lyon 1, Univ-Lyon, CNRS, UMR 5223, 15 Bd Latarjet, CEDEX, 69622 Villeurbanne, France; 3Centre de Recherche, Hutchinson, Rue Gustave Nourry—B.P. 31, 45120 Chalette-sur-Loing, France

**Keywords:** extrusion foaming, physical foaming, TPV, modelling

## Abstract

This work focuses on the extrusion foaming under CO_2_ of commercial TPV and how the process influences the final morphology of the foam. Moreover, numerical modelling of the cell growth of the extrusion foaming is developed. The results show how a precise control on the saturation pressure, die geometry, temperature and nucleation can provide a homogeneous foam having a low density (<500 kg/m^3^). This work demonstrates that an optimum of CO_2_ content must be determined to control the coalescence phenomenon that appears for high levels of CO_2_. This is explained by longer residence times in the die (time of growth under confinement) and an early nucleation (expansion on the die destabilizes the polymer flow). Finally, this work proposes a model to predict the influence of CO_2_ on the flow (plasticizing effect) and a global model to simulate the extrusion process and foaming inside and outside the die. For well-chosen nucleation parameters, the model predicts the final mean radius of the cell foam as well as final foam density.

## 1. Introduction

In order to reduce the consumption of plastic, weight reduction by foaming is increasingly used. To allow such performance without environmental disagreement, physical foaming by harmless gases (CO_2_ or N_2_) is a preferred and suitable solution. Indeed, it allows the foaming of the polymer without any production of harmful gas and does not pollute the polymer [1,2].

Physical foaming is obtained by variations in temperature or pressure which create local supersaturation of CO_2_. To generate these thermodynamic instabilities, a continuous extrusion process is widely used both on a laboratory scale and on an industrial scale. This process uses a high depressurisation rate to foam (depressurisation due to the geometry of the die). Moreover, this process has been proved to be effective on polymer blends so as to control their density while having acceptable mechanical properties [3].

A thermoplastic vulcanizate (TPV) is made of a thermoplastic matrix filled with dynamically crosslinked rubber domains [1,4]. It exhibits vulcanized rubber-like mechanical properties while being reprocessable and recyclable. TPV are polymer materials of interest to create high added value foams. Their TPV extrusion foaming [1,5,6] has already been investigated on an industrial reference (Santoprene grades) demonstrating that TPV foaming can be achieved using various foaming agents. It has been shown that the use of N_2_ leads to smaller cell sizes, explained by its high nucleation capacity, and CO_2_ gives the lowest foam density, explained by its high solubility [6,7]. So, the CO_2_ is the physical blowing agent (PBA) of choice (as a green blowing agent) for TPV. Based on the literature for TPV foaming, only the thermoplastic phase can expand due to the high level of crosslinking content of the EPDM phase [1,5,6].

Therefore, based on PP extrusion foaming literature, it has been demonstrated that a high depressurisation level increases the rate of nucleation [7] and induces a better foam morphology (homogenous cells and low density < 100 kg/m^3^) [8,9]. It has also been demonstrated that the temperature influences the foaming of PP [10] (and so the foaming phase of the TPV). For example, a decrease in temperature induces a decrease in cell diameter. An optimum of temperature exists depending on the polymer and its characteristic temperature (T_m_,T_g_) [11].

The numerical modelling of TPV foaming has not been investigated but there is a need due to the growing interest on TPV foaming. The interest in the modelling of the process is increasing as well. Moreover, it can enable predictability of the resulting foam density and morphology.

## 2. Materials and Methods

The industrial TPV (polypropylene thermoplastic phase and EPDM crosslinked rubber phase filled with carbon black) Vegaprene 528 was provided by Hutchinson (45 120 Châlette-sur-Loing, France).

Batch foaming measurement were made on a Paar Instrument mini bench reactor (4567). Temperature was controlled by the Paar system and CO_2_ was fed. Pressure was controlled by a Linde pump (DSD 500) which allowed for its regulation so as to reach maximal working pressure of 34.5 MPa. The precision of the pump is about 0.1 MPa and enable mass flow rate of CO_2_ up to 5 kg/h. The depressurisation rate was set manually to 1.5 MPa/s. The saturation time was set to 45 min at 180 °C, then the foaming temperature was set and a delay of 45 min at equilibrium was set before quenching. This results in a total process time of about 2 h.

The extrusion foaming stage was carried out with a twin-screw extruder, that enabled to control each parameter separately (pressure, mass flow rate, temperature (die and extruder separately). The extruder used was the BC 21 (L/D = 36, screw diameter = 25 mm) from Clextral using in co-rotative configuration. The inlet of CO_2_ is shown in Figure 1 and the upstream CO_2_ sealing was made through screw profile and downstream by the gear pump. Due to the configuration of the experimental device, the CO_2_ concentration and the solubility limit are equivalent on the extruder. The CO_2_ is injected into a partially filled area of the extruder and the pump maintains a constant saturation pressure by adjusting the CO_2_ flow rate. Due to the small difference between the temperature on the die and on the extruder, it is assumed this is always verified (temperature on the extruder 180 °C). The schematic expected pressure profile on the extruder is demonstrated in Figure 2. The rotational speed was set at 200 rpm and the temperature of the sleeve set at 180 °C (saturation pressure).

Due to technical limitations of the system, the lowest pressure of CO_2_ that can be set (at 160 °C) is 5 MPa (for the die having a diameter of 1.5 mm); a lower value led to an increase of the pressure applied on the extruder beyond the safety point.


**Modelling**



**Cell growth Simulation**


In order to model foam growth, three equations have to be resolved [3,12,13,14]: Equation (1) is the balance of forces (gas pressure, viscosity, surface tension forces); Equation (2) expresses the mass conservation of the dissolved species and the diffusion between polymer and cell; and Equation (3) corresponds to the diffusion of the gas through the outside layer of the cell.
(1)$$P_{\mathrm{bub}}-P_{\mathrm{sys}}-\frac{2\gamma }{R}+2\int_{R\left(t,t^{\prime }\right)}^{R_{\sup\left(t,t^{\prime }\right)}}\frac{\tau_{\mathrm{rr}}-\tau_{\theta \theta }}{r}\mathrm{dr}=0$$
(2)$$\frac{d}{\mathrm{dt}}\left(\frac{{4\pi P}_{D}R^{3}}{{3R}_{g}T}\right){=4\pi R}^{2}D{\frac{\partial c}{\partial r}}_{r=R}$$
(3)$$\frac{\partial c}{\partial t}+\frac{\dot{R}R^{2}}{r^{2}}\frac{\partial c}{\partial r}=\frac{D}{r^{2}}\frac{\partial }{\partial r}\left(r^{2}\frac{\partial c}{\partial r}\right)$$

P_bub_ is the pressure on the bubble, P_sys_ is the pressure outside the polymer, R is the radius of the polymer, γ is the interfacial tension of the polymer, $\tau_{\mathrm{rr}},\tau_{\theta \theta }$ are the local stress around the cell, R_sup_ is the radius of the influenced volume.

D is the diffusion coefficient, T the temperature, R_g_ the gas constant, c the concentration of CO_2_, r is the radius of the cell as a function of time.

For complex fluids such as polymers, the stress component (Equation (1)) has to be expressed depending on strain rate. The constitutive equation used is mostly the upper convective Maxwell model (Equations (4) and (5)) [15] to predict the viscoelastic behaviour of the polymer during cell growth. The relaxation time spectrum is extracted from dynamic shear rheology at multiples temperature (160, 170, 180 °C) in order to model the relaxation time spectrum as a function of the temperature The initial R_sup_ is fixed using experimental data as an adjustable parameter.
(4)$$\frac{d\tau_{\mathrm{rr}}}{\mathrm{dt}}=-\left(\frac{1}{\lambda \left(T\right)}+\frac{4\dot{R}R^{2}}{r^{3}}\right)\tau_{\mathrm{rr}}-\frac{4\eta \left(T^{{}^{\circ}}\right)\dot{R}R^{2}}{\lambda \;\left(T\right)r^{3}}$$
(5)$$\frac{d\tau_{\theta \theta }}{\mathrm{dt}}=-\left(\frac{1}{\lambda \left(T\right)}-2\frac{\dot{R}R^{2}}{r^{3}}\right)\tau_{\theta \theta }+2\frac{\eta \left(T^{{}^{\circ}}\right)\dot{R}R^{2}}{\lambda \;\left(T\right)r^{3}}$$

$\tau_{\mathrm{rr}}$ and $\tau_{\theta \theta }$ are the local stress around the cell, R is the radius at time t, $\dot{R}$ the expansion rate of the radius at time t, η and λ are the viscoelastic parameter of the polymer.

In order to take into account, the CO_2_ plasticizing effect on the rheological properties, a model developed in our previous work [16] is used as will be developed later. To model the influence of cells among themselves, a non-influence volume is assumed [13]; it is defined as the volume of polymer which is not impacted by the presence of a neighbouring cell. The concentration inside the volume of non-influence is the same at each point. As a result, the non-influence volume defines the volume where nuclei can appear. The nucleation stops when the concentration of CO_2_ on this volume reaches a given value defined as the concentration for a nucleation rate equal to 0.01*the initial value of nucleation rate. As the nucleation stops, each cell has a specific volume to grow (influence volume of each cell). This hypothesis enables to take into account the interaction from cell to cell without taking into account coalescence.

In this work, in terms of rheology and diffusivity, the material is assumed to be homogeneous due to the similarity of PP and EPDM in terms of CO_2_ diffusion and solubility. Since the dispersion of the EPDM/filler is fine enough regarding the characteristic volume influencing the growth (radius ≈ 10 µm, radius of EPDM nodule ≈ 500 nm), the mechanical behaviour of the polymer is assumed to be homogeneous.


**Nucleation simulation**


Instead of the classical homogeneous nucleation (Equation (6)), the nucleation model that has been chosen is the one defined by Park et al. [17] which takes into account the material heterogeneity by assuming a contact angle (θ) (adjustable parameter) and setting a randomly disperse geometries surface (β between 0° to 90° with a medium value of 45°). Then, the modified nucleation rate is described by Equations (7) and (8). However, the heterogeneity induced by PP crystallization, EPDM nodules and carbon black filler (in our material) are not taken into account. Therefore, the nucleation rate as defined by Equations (7) and (8) depends on the thermodynamic variables and the polymers’ characteristics.
(6)$$J_{\mathrm{Hom}}=N\sqrt{\frac{{2\gamma }_{\mathrm{lg}}}{M\pi }}\exp\left(\frac{{-16\pi \gamma }_{\mathrm{lg}}^{3}}{{3K}_{B}T{\left(P_{\mathrm{Bub}}{-P}_{\mathrm{Sys}}\right)}^{2}}\right)$$
(7)$$\begin{array}{l} J_{\mathrm{Hom}\;\mathrm{corrected}}\\ =\int_{0^{{}^{\circ}}}^{90^{{}^{\circ}}}\frac{N^{\frac{2}{3}}Q\left(\theta_{C},\beta \right)e^{-\frac{{\left(\beta -\bar{\beta }\right)}^{2}}{2\sigma^{2}}}}{\sigma \sqrt{2\pi }}\sqrt{\frac{2\gamma_{\mathrm{lg}}}{\pi \mathrm{mF}\left(\theta_{C},\beta \right)}}\exp\left(-\frac{16{\pi \gamma }_{\mathrm{lg}}^{3}F\left(\theta_{C},\beta \right)}{3k_{B}T{\left(P_{\mathrm{bub}}-P_{\mathrm{sys}}\right)}^{2}}\right)d\beta \end{array}$$
(8)$$F\left(\theta_{C}\right)=\frac{2+3{\cos\theta }_{C}-\cos^{3}\theta_{C}}{4}$$

N is the total number of CO_2_ molecules (possible site of nucleation), $\gamma_{\mathrm{lg}}$ is the surface tension of the polymer, M is the molar mass of the CO_2_, k_B_ the Boltzmann constant, T the temperature, P_sys_ the pressure of the system and P_Bub_ the saturation pressure. F corresponds to the heterogeneity function.

**Plasticizing model**:

The plasticizing model used in this work to take into account the influence of CO_2_ solubilization on the viscosity of the polymer (shear and extensional) is the one developed in our previous work [16]. The plasticizing influence of CO_2_ depends on the plasticizer volumetric fraction (φ) that is derived from Daoud et al. [18] and that expresses the viscosity shift factor:(9)$$A_{C}\;\alpha \;{\left(1-\varphi \right)}^{4}$$

To use this model (Equation (9)), the volumetric fraction of CO_2_ has to be calculated since the additivity of volume is not validated for gas/polymer mixtures. The equation of state of Sanchez Lacombe [19,20] (Equation (10)) is used to calculate the actual volumetric fraction of CO_2_ (Equation (11)). The volumetric fraction that is considered is the one at pure state $\varphi_{i}^{0}$.
(10)$$\tilde{P}=-{\tilde{\rho }}^{2}-\tilde{T}\left(\ln\left(1\;-\tilde{\rho }\right)+\left(1-\frac{1}{r}\right)\tilde{\rho }\right);\;r=\frac{{\mathrm{MP}}^{*}}{{\mathrm{RT}}^{*}\rho^{*}}$$

$\tilde{P},\;\tilde{\rho },\;\tilde{T}$ are the reduced pressure, density, and temperature, respectively, and M is the molar mass.
(11)$$\varphi_{i}^{0}=\frac{\frac{\varphi_{i}P_{i}^{*}}{T_{i}^{*}}}{\sum_{j}\frac{\varphi_{j}P_{j}^{*}}{T_{j}^{*}}}$$

P*, T* are the critical pressure and temperature, $\varphi_{j}$ is the volumetric fraction of phase j.


**Foam growth resolution**


To model the foaming process in the extruder, die, pressure, flow and polymer temperature are first calculated by coupling the Navier-Stokes and thermal equations using the finite element method under COMSOL^®^ as explained in Figure 3. Then the foaming modelling uses the traditionally used equations that are specified hereafter (Equations (1)–(5)). The influence of foaming on extrusion modelling is considered using the variation of volume induced by cell growth. As a result, the foaming inside the die, through a loop, modifies the characteristic of the flow (flow rate, pressure), and thus the onset of nucleation and the behaviour inside the die. It has been evaluated that after 5 loops, a convergence to a unique solution occurs (nucleation onset on the die).

Figure 1 shows the principle of the numerical modelling of the extrusion foaming. On the right side, the new input is specified at each step. The stress in Equation (1), calculated using the upper convective Maxwell model (Equations (4) and (5)), is based on the relaxation spectrum extracted from dynamic shear rheology. The cells are supposed to be purely spherical, cell-cell interactions are neglected, cell coalescence is not considered and the gas diffusion towards the environment is not taken into consideration.

## 3. Results and Discussion

### 3.1. Influence of CO_2_ Content on Foaming Behaviour

To study the foamability of the TPV, a first batch foaming session was conducted to determine the optimum foaming temperature and pressure. It was demonstrated that the TPV does not foam properly between 130 and 170 °C with a pressure in the range of 2–8 MPa. We concluded that the depressurization rate was too low and that the nucleation could not compete with cell growth. As a result, coalescence tended to appear.

The scheme of the die and the depressurization rate induced by the die system for 3 different saturation pressures is shown in Figure 4. In order to verify the accuracy of the plasticizing model developed in our previous work [16] taking into account the effect of temperature on the PP phase, the experimental pressure at the inlet of the die was compared with the computed one using FEM (finite element method) (COMSOL). Figure 4 shows that the present model (Equations (9)–(11) can predict quite accurately the plasticizing effect of CO_2_ on the flow inside the die.

According to the literature, increasing the content of CO_2_ implies a nucleation upstream on the die (increase of CO_2_ induces early nucleation) [21], combined with the decrease of viscosity induced by CO_2_ (increase of cell growth rate). Thus, these two phenomena can induce a variation of the growth time on the matrix and create a coalescence (as we observed during batch foaming). Furthermore, this phenomenon might by emphasised due to the shear of the cell produced by the flow. In the literature, it has been demonstrated that a late nucleation induces better foam morphology [13]. This is confirmed by the SEM observations of the foam morphology as shown in Figure 5. Lower amounts of CO_2_ lead to better foamability (no coalescence, no flow instabilities) and lower final foam densities. This is due to the onset of nucleation and the lowest growth time inside the die, which resulted in few or no coalescence. Indeed, counterintuitively the increase of CO_2_ content does not lead to a decrease of density, demonstrating a limit to the maximal expansion of the TPV, due to the early nucleation and coalescence in the die.

It is postulated in the literature that only the foaming of the thermoplastic phase of TPV is possible due to the crosslinking of the EPDM phase [2]. For the lowest achieved density (480 kg/m^3^), if only the PP phase has expanded, the equivalent density of the PP phase is about 250 kg/m^3^ which can be achieved for pure PP [10]. To verify this hypothesis, TEM observations were performed on the TPV samples (Figure 5a) with the lowest density. In Figure 6a,c, the TEM image shows the presence of EPDM and PP in the cell wall which indicates the involvement of the two phases during foaming. In Figure 6, the cell wall shows a PP/EPDM blend at the surface, which indicates an expansion of the PP phase then merging with the EPDM phase described in Figure 7. The presence of multiple cells induces EPDM deformation due to the confinement induced by the cells.

The two-step foaming of the TPV: we observe the foaming of PP at first then, as the cell expands, the EPDM nodules move through the PP phase, acting only as a classical filler. In conclusion, the EPDM nodules act as a reinforcement charge during the foaming of the PP phase. Besides, due to the limitations of the 2D TEM observation, no conclusion can be made on the influence of the carbon black filler, even though it seems unlikely to be acting as a nucleating agent.

In order to study the influence of saturation pressure and depressurization rate on the density of the final foam, another die was designed, the difference of pressure inside both dies is shown in Figure 8. The inlet pressure is similar but generates a different rate of depressurization (Table 1) almost 4 times as high as for Die 1 compared to the second one. Only the density is studied and compared depending on the content of CO_2_. In Figure 9, the die having the highest drop of pressure and the lowest time of depressurisation enables better expansion of the TPV as demonstrated on Figure 9. It also appears that for each die, a lower value of CO_2_ pressure enables a lower density which is directly related to a lower time of growth on the die and nucleation closer to the outlet.

### 3.2. Numerical Modelling of Cell Growth

We assumed an onset of nucleation at a difference of pressure of 1.5 MPa (which induces a nucleation rate of 1 × 10^12^ cell·s^−1^ m^−3^, which has been proven to be the nucleation rate at the onset of nucleation for PP [7,21]). In addition, for pressures under 1.5 MPa of CO_2_ saturation, no nucleation appears, which confirms the minimal difference of pressure of 1.5 MPa. The maximal time of growth outside the die is defined as the time of cooling outside due to the natural air convection. After a loss of 15 °C, the growth (temperature of 145 °C close to the crystallization temperature of the PP phase) is assumed to be stopped. The simulation also stops if the total initial amount of gas is used during cell growth. Using thermal modelling with only natural convection, the time of cooling outside the die was calculated at 3 s using COMSOL thermal modelling. The maximum time of growth was therefore set at 3 s.

The initial parameters are listed in Table 2. The superior radius is defined in the literature as the volume that is mechanically influenced by the cells [13]. To determine the value of the superior radius, the modelling of the uniform and homogeneous cell structure obtained at 160 °C and 5 MPa using Die 1 for various value of R_Sup_ was conducted and compared to the distribution of the TPV foamed (Figure 9).

The optimum value of R_Sup_ was evaluated at 11 µm which induces a mean radius of 31 µm (Figure 10). In addition, the total number of nuclei at the end of modelling was about 1 × 10^13^ cell/m^3^ (calculated using J_hom corrected_ from Equation (7)) at the end of the nucleation phase; compared to the experimental number of nuclei of 9 × 10^12^ cell/m^3^, it shows accordance between modelling and experimental data. This demonstrates the ability of Park et al.’s model [3] to predict the total number of cells. In addition, the coupling with COMSOL enables a modelling of the expansion of the TPV during extrusion. The density of the modelled extrusion foaming at a saturation of 5 MPa was 510 kg·m^−3^ (for R_sup_ set at 11 µm) compared to the effective density of 480 kg·m^−3^. This demonstrates the ability of the model to predict TPV foaming. The success of the modelling depends on the value of R_sup_ as well as on the model of nucleation chosen; in our case the modified nucleation model shows great predictability at 5 MPa.

The modelling of the extrusion foaming stage at 160 °C saturated at 5 MPa demonstrated a great concordance with experimental, as the R_Sup_ is set at 11 µm and so the R_Sup_ is then fixed for each other modelized extrusion foaming condition.

In order to verify the ability of our model to predict the foaming behaviour of the TPV extrusion, the modelling of lower saturation values with a different die (which enables lower saturation pressure at 160 °C) was conducted. The saturation pressure was set at 4 MPa and another set of parameters was used: 165 °C and 3 MPa of saturation. The resulted density for each model is shown in Table 3 and demonstrates the accuracy of our model to predict the foaming behaviour of complex polymers (TPV) for low saturated pressures. The modelling at 165 °C induces variation of nucleation and cell growth due to the dependence on the temperature of the cell growth rate and nucleation rate.

However, the modelling for TPV saturated at 6 MPa (R_sup_ fixed at 11 µm) demonstrated no perfect accordance between experimental and modelling: the final modelling density is about 410 kg/m^3^ when the experimental one is about 610 kg/m^3^. This is explained by a longer growth time inside the die which could induce higher coalescence (which is considered as null in our model). Furthermore, a non-isotropic morphology is observed (Figure 5b) due the orientation of the cells under the flow, so that cell stability is no longer validated at 6 MPa and higher pressures, which explains the difference between experimental and modelling. From an experimental point of view, the actual limitation is the process instability itself due to the flow dominance in the die and its effect on cell growth. The experimental limitation of 6 MPa (appearance of flow instabilities) is also the divergence between experimental result and our model, which means that our model works well for stable flows. Furthermore, the influence of shear stress on the nucleation rate needs to be investigated to improve the understanding of the nucleation phase.

## 4. Conclusions

The extrusion foaming of TPV was successfully conducted in order to achieve a controlled density reduction from 900 to 480 kg/m^3^ depending on saturation pressure and die temperature. However, for high values of CO_2_ saturation pressure (*p* > 6 MPa), foam instabilities occur mainly due to the early nucleation inside the die and cell growth under confinement. An optimum of properties was determined that enables a density around 480 kg/m^3^ with a homogeneous and isotropic cell morphology.

The application of cell growth models coupled with finite element methods (flow and thermal equations) on the extrusion foaming on TPV was successfully set. Our model demonstrated great accuracy to predict experimental foam densities for different values of temperature and pressure using only one experimental data used as fitting parameter and determined from one experimental processing condition. This model serves as a predicting modelling tool to optimize the experimental conditions and die geometry to achieve a chosen density and for a wanted mean diameter.

The only limitation of our model comes from the experimental limitation due to the flow instability observed at *p* > 6 MPa and an improvement of the model could be derived by considering flow instabilities and how it affects cell-cell behaviours such as coalescence.

## Figures and Tables

**Figure 1 polymers-14-04513-f001:**
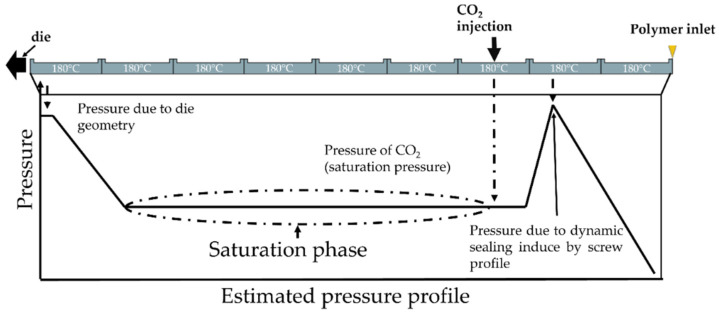
Scheme of the pressure profile on the extruder and area of CO_2_ inlet.

**Figure 2 polymers-14-04513-f002:**
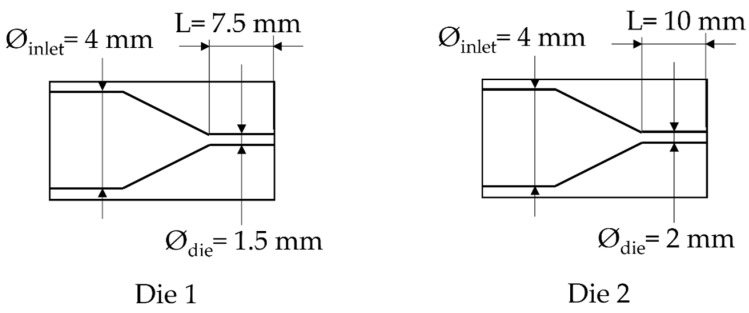
Scheme of the die used.

**Figure 3 polymers-14-04513-f003:**
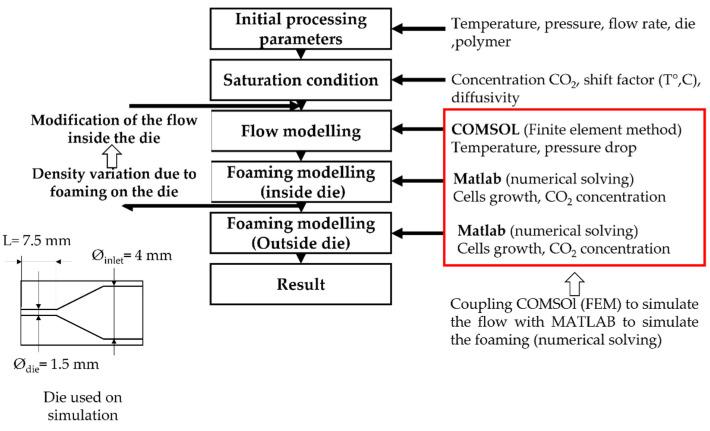
Scheme of the principal modelling of TPV extrusion foaming.

**Figure 4 polymers-14-04513-f004:**
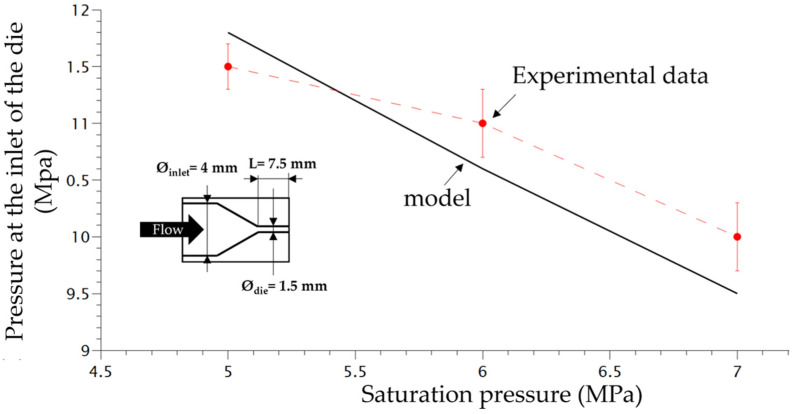
Comparison between experimental pressure at the inlet of the die with the model pressure using FEM method (Comsol).

**Figure 5 polymers-14-04513-f005:**
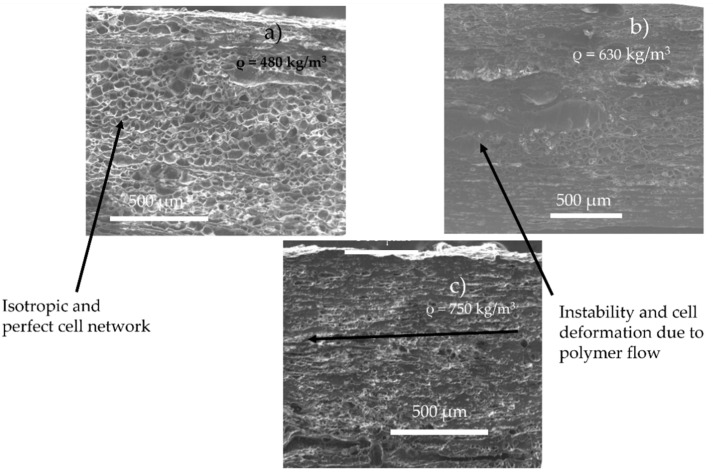
Foam morphology and density for TPV foam made by extrusion foaming at various value of CO_2_ pressure (die diameter = 1.5 mm, length 7.5 mm) at 160 °C and 5 MPa (**a**), 6 MPa (**b**), 7 MPa (**c**).

**Figure 6 polymers-14-04513-f006:**
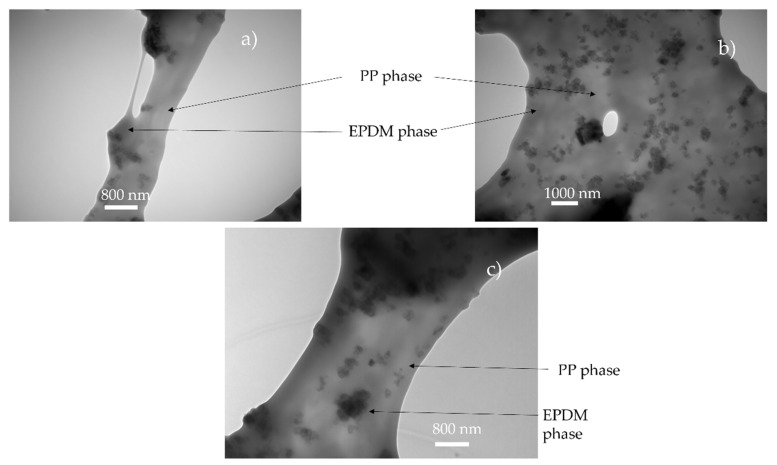
TEM observation of foam obtain for saturation value of 5 MPa and 160 °C (Figure 5a–c: (**a**) 5 MPa, (**b**) 6 MPa, (**c**) 7 MPa). Due to the carbon black in the EPDM, the EPDM phase appears darker than PP.

**Figure 7 polymers-14-04513-f007:**
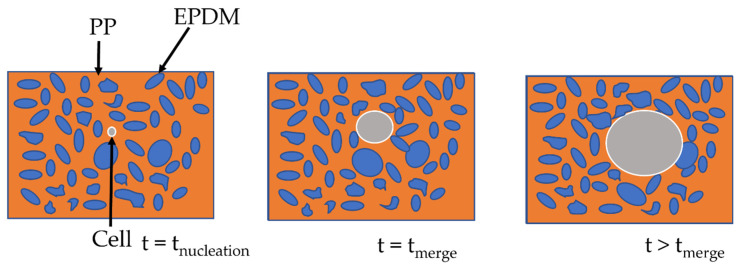
Scheme of the cell growth on TPV.

**Figure 8 polymers-14-04513-f008:**
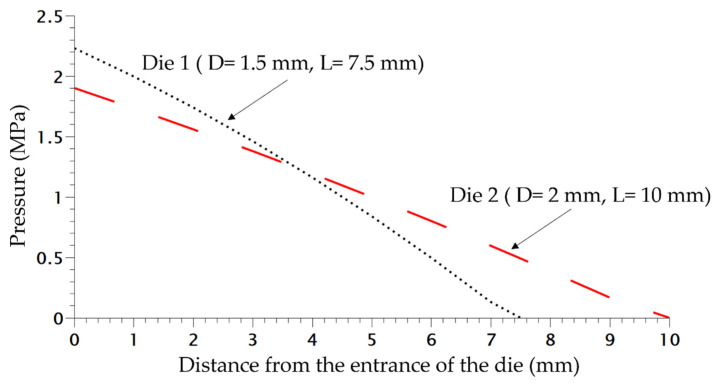
Pressure variation in both dies at 220 °C without CO_2_.

**Figure 9 polymers-14-04513-f009:**
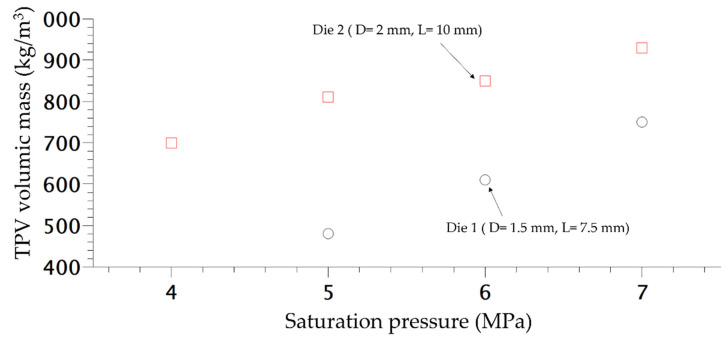
Density variation of extruded foam produced at 160 °C depending of the CO_2_ pressure.

**Figure 10 polymers-14-04513-f010:**
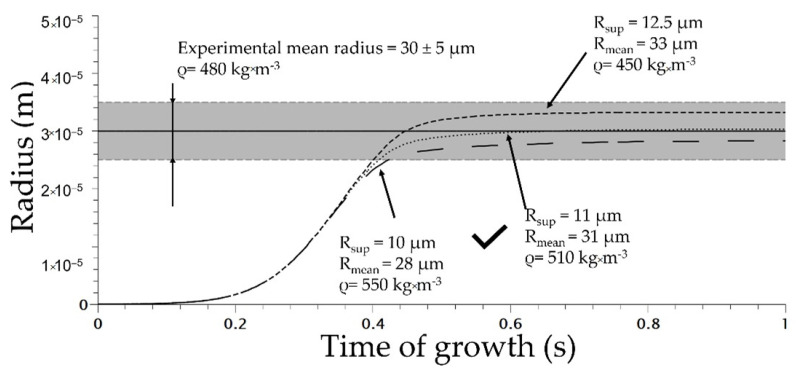
Cell growth modelling on extrusion foaming process for various superior radii (die used 1 on Figure 2).

**Table 1 polymers-14-04513-t001:** Characteristic of the dies and associated depressurization variation.

Die	Radius (mm)	Length (mm)	Drop Pressure (MPa) ^1^	Drop Rate Pressure (MPa/s)
1	0.75	7.5	9.7	135
2	1	10	9.2	45

^1^ 220 °C, without CO_2_.

**Table 2 polymers-14-04513-t002:** Parameters of extrusion foaming computation.

Parameters	Value
Range of R_sup_	Between 10 and 12.5 µm
K_H_ (at 433 K)	1.64 × 10^−4^ mol Pa/m^3^ [11]
D (at 433 K)	6 × 10^−9^ m^2^/s [2]
R_0_	38 nm
Mass flow rate	2 kg·h^−1^

**Table 3 polymers-14-04513-t003:** Result of TPV extrusion foaming modelling compared to experimental data.

Saturation Pressure (MPa)	Die	T (°C)	Mean Radius Experimental (µm)	Mean Modelized Radius (µm)	Experimental Density (kg/m^3^)	Modelized Density (kg/m^3^)	Modelized Number of Cells (Cells/m^3^)	Experimental Number of Cells (Cells/m^3^)
7	1	160	Deformed cell	12	750	520	8.5 × 10^13^	None
6	1	160	Deformed cell	26	610	410	1.6 × 10^13^	None
5	1	160	30	31	480	510	8 × 10^12^	8 × 10^12^
4	2	160	25	27	700	650	6 × 10^12^	5 × 10^12^
3	2	165	19	22	770	740	4 × 10^12^	2.5 × 10^12^

## Data Availability

Not applicable.

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
