# Peer review of "TPV Foaming by CO_2_ Extrusion: Processing and Modelling"

_polymers, 2022, doi:10.3390/polym14214513_

Round 1
Author Response
Dear reviewer,
We thank you for the very thorough reading of our manuscript. The various corrections have been made in the text and the various discussion points have been answered in a separate file.
Best regards
Y. Chalamet

Reviewer 2 Report
The work provided one model to predict the influence of CO2 solubilization on the flow (plasticizing effect) and a global model to simulate the extrusion process and foaming inside and outside the die. Generally, the study presented one new method, which would simplify the foaming process for high levels of CO2 . However, there are bigger error on the model accuracy. Because the final modelling density is about 410 kg/m3 when the experimental one is about 610 kg/m3 . I suggest that the work should improve the accuracy.
Author Response
Dear reviewer,
We thank you for the very thorough reading of our manuscript. The various corrections have been made in the text.
Best regards
Y. Chalamet
Round 2
Reviewer 1 Report
See the reviewer's comments in red in the word file.

Author Response
Dear Colleague,
We thank you for your careful reading of our manuscript. We have incorporated all your comments into the manuscript that we are submitting.
Best regards
Reviewer 2 Report
I agree that the revised manuscript can be accepted in the journal.
Author Response
Dear Colleague,
We thank you for your careful reading of our manuscript.
Best regards